# Poor sleep quality and its associated factors among pregnant women in Northern Ethiopia, 2020: A cross sectional study

**Tadeg Jemere**[1]*, **Berhanu Getahun**[2], **Fitalew Tadele**[1], **Belayneh Kefale**[3], **Gashaw Walle**[1]

**1** Department of Biomedical Sciences, College of Health Sciences, Debre Tabor University, Debre Tabor, Amhara, Ethiopia, **2** Integrated Emergency Surgery and Gyn/Obs Professional Specialist, Amdework Primary Hospital, Amdework, Wollo, Ethiopia, **3** Clinical Pharmacy Unit and Research Team, Department of Pharmacy, College of Health Sciences, Debre Tabor University, Debre Tabor, Amhara, Ethiopia

* tadegamare12@gmail.com

## Abstract

### Background

Sleep is a physiologic necessity for all humankind. Pregnant women, in particular, need adequate sleep to develop their fetuses as well as save energy required for delivery. A change in sleep quality and quantity is the most common phenomena during pregnancy due to mechanical and hormonal factors. However, there is a scarcity of data about poor sleep quality and its associated factors among pregnant mothers in Ethiopia. Therefore, this study aims to determine the prevalence of poor sleep quality and its associated factors among pregnant mothers at Wadila primary hospital, Ethiopia.

### Methods

Institution based cross-sectional study design was employed on 411 pregnant mothers. Data were collected using a pre-tested interviewer administered questionnaire. SPSS Version 23 for Windows software was used for data analyses. Bivariate analysis was conducted to detect the association between dependent and independent variables, and to choose candidate variables (p < 0.25) for multivariate logistic regression. Statistical significance was set at p-value <0.05.

### Results

A total of 411 participants were included in the study making a response rate of 97.4%. Overall, 68.4% of participants found to have poor sleep quality (PSQI>5). Age of the mother [age 20–30 years; AOR = 4.3 CI (1.8, 9.9), *p* = 0.001, and age >30 years; AOR = 4.7 CI (1.6, 13.9) *p* = 0.005], gestational age [second trimester, AOR = 2.46, CI (1.2, 4.9), p = 0.01 and third trimester, AOR = 7.5, CI (3.2, 17.8), p = 0.000] and parity [multiparous women; AOR = 2.1(1.24, 3.6) p = 0.006] were predictor variables for poor sleep quality among pregnant mothers.

**Data Availability Statement:** Data are all contained within the manuscript.

**Funding:** The authors received no specific funding for this study.

**Competing interests:** The authors have no competing interests to declare.

## Conclusion

More than two-third of pregnant mothers had poor sleep quality. Advanced maternal age, increased gestational age and multiparty are found to be predictors of poor sleep quality in pregnant women.

## Introduction

Sleep is a physiologic necessity for humans and is essential for the normal functioning of the body. A good sleep is fundamental for a healthy pregnancy and used to conserve energy for delivery process [1, 2]. A change in sleep quality and quantity is a common phenomenon during pregnancy. The total sleep time varies through pregnancy, with an increased total sleep time in the 1st trimester of pregnancy, normalized sleep time in the 2nd trimester and decreased sleep time in the 3rd trimester. More frequent or prolonged awakenings are observed with increasing gestational age [3, 4].

Sleep can be altered during pregnancy due to mechanical and hormonal factors. This may be due to physical and mental changes that the body undergoes during pregnancy. Problems such as leg cramps, urinary incontinence, shortness of breath, heart burn, and intense backaches are among the mechanical factors. Some women may also have difficulty in finding the right position to sleep because of the growing uterus, a very active baby, and worries about the baby and themselves [4, 5]. There is also a marked increase in the level of estrogen and progesterone during pregnancy. Increasing evidence suggests that, these hormones control reproductive functions and influence a diverse range of physiological and psychological processes, including sleep, mood, sensory, and cognitive functions [6]. The level of estrogen is elevated more than 100 times and progesterone up to 200 times in late pregnancy compared to pre-pregnancy level. Estrogen causes edema of the upper airway mucosa, leading to increased upper airway resistance and difficulty in breathing. Oxytocin, the hormone responsible for uterine contractions, is known to peak at night, possibly causing sleep fragmentation in late pregnancy [6, 7].

Prevalence rates for altered sleep quality during pregnancy range from 40% to 96%, depending on the population studied and the time of assessment being higher during the third trimester of pregnancy [8–12]. According to the result of different studies, increased age, advanced education, employment status, monthly income, gestational age, parity, alcohol consumption and smoking are found to be factors affecting the sleep quality of pregnant women [4, 13–15].

Poor sleep quality may bring about mental impairment of the mother and fetal neural networks are highly vulnerable to maternal sleep loss. The offspring are predisposed to various anxiety disorders and learning disabilities [9]. Though the prevalence and factors affecting poor sleep quality among pregnant mothers have been investigated in developed world, this is not well understood in developing countries like Ethiopia. Hence, this study aims to determine the prevalence of poor sleep quality and its associated factors among pregnant mothers.

## Methods and materials

### Study setting and period

Institution based cross-sectional study design was employed at Wadila primary hospital, Kone town. Kone town is located in Amhara regional state, North Wollo zone, 623 kilometers north of Addis Ababa, Ethiopia. Wadila primary hospital is one of the youngest hospitals in Ethiopia

and providing service for approximately 200,000 people in the catchment area, including antenatal care (ANC) service for pregnant mothers. The data were collected from May 15 to August 15, 2020.

## Study population

All pregnant mothers who came to the hospital for ANC follow up and present at the time data collection were study populations.

## Sample size determination and sampling procedure

A single population-proportion formula was used to calculate the sample size by considering the proportion of poor sleep quality as 50% (since no study is conducted in Ethiopia), with a 95% confidence interval, 5% margin of error, and with the assumption of 10% non-response rate. Hence, the final sample size was 422. Study participants were selected using systematic random sampling technique. The average daily ANC follow up of pregnant mothers in the hospital was 18, making a total 1188 study population during the study period. By dividing the study population to sample size, we got, a K value of 2.8≈3. To get the required sample size every other pregnant mother was selected.

## Eligibility criteria

All pregnant mothers who came to the hospital for ANC follow up and present at the time of data collection were included. Whereas, pregnant mothers who had established psychiatric disorder, chronic diseases and who work in night shifts in the last month were excluded.

## Data collection and management

Data were collected using a pre-tested interviewer administered questionnaire which contains four parts. These includes: sociodemographic and substance use-related variables, medical record reviews and pregnancy-related variables, a validated standardized test for sleep quality, the Pittsburgh Sleep Quality Index (PSQI) (16), and blood pressure (BP) measurement.

Blood pressure was measured using an automated digital BP monitor complete with adult cuff and participants in sitting position with the arm at the heart level and after 5 minutes rest.

The PSQI is a 19-item self-report measure designed to measure sleep quality over the past month. It has seven subscales and these subscales are added to determine a global sleep quality score (GSQ). The GSQ score ranges from 0 to 21, in which higher scores (PSQI >5) indicate poor sleep quality (16).

The data were collected by trained midwives. A week before the actual time of data collection, the questionnaire was tested on 5% of the participants having similar sociocultural characteristics with the study participants at Kone health center.

## Operational definitions

Good sleep quality is a state of having PSQI score ≤5 and individuals who had PSQI score >5 were said to have poor sleep quality [16]. Participants who consumed psycho-stimulant substances such as alcohol, khat, and cigarette smoking at least once within the last 30 days were classified as current user.

## Data analysis

After checking its completeness, the collected data were coded then entered into Epi-data version 3.1 and exported to SPSS Version 23 for Windows software. Frequency, mean and proportions

were used for the descriptive analysis of data. Bivariate analysis was conducted to detect the association between dependent and independent variables, and to choose candidate variables ($p < 0.25$) for multivariate logistic regression. We used $p < 0.25$ as a cutoff point to choose candidate variables of the last model so as to improve the chances of holding significant confounders. Odds ratio and its 95% confidence interval were assessed for potential indicators of poor sleep quality, which were included in the last model. Statistical significance was set at p-value $<0.05$.

### Data quality management

To guarantee the quality of data, pre-tested interviewer administered questionnaire was utilized. A 2 day training was given for data collectors regarding the purpose of the study and measurement techniques. The questionnaire was translated to Amharic language and then retranslated back to English to preserve its consistency. To decrease bias, participants were assured to keep their response confidential.

### Ethics approval and consent to participate

Ethical approval was obtained from ethical review Committee of Debre Tabor University with ethical approval number of DTU/4/258/2020 and from Ethical review Committee of Wadila primary hospital with Ethical approval number of WPH/1/1/2020. Written informed consent was taken from each study participants. Confidentiality of information was kept properly.

## Results

### Sociodemographic characteristics

A total of 411 participants were included in the study making a response rate of 97.4%. The majorities (71.3%) were in the age range of 20–30 years with the mean age of 25.7 ± 5.5. Around 30% of participants did not attend formal education and 28.2% attend elementary education. Majorities (94.9%) were married and the lowest percentages were widowed (1.7%). More than half (52.8%) were housewives and 19% of them were government employees. From all participants, 52.1% were rural residents and the average monthly income of the participants was 1775.7 ± 1610 (Table 1).

### Pregnancy related characteristics

More than half (54.3%) of the participants were in the second trimester of pregnancy and the lowest percentages (12.2%) were in the first trimester of pregnancy. On the other hand, 65.5% of participants were multiparous and the rest 34.5% were nulliparous. From all participants, only 4.1% had history of still birth and 7.5% had history of abortion. The majority (76.9%) of participants received iron and folic acid supplementation. Regarding their blood pressure, 6.1% of participants had increased blood pressure at the time of data collection.

### Substance use profile of study subjects

As per the current study result, all participants (100%) did not chew khat and smoke cigarette even at least once in their life time. Regarding alcohol consumption, (37.2%) of participants found to have history of alcohol consumption and 17% of them currently practiced drinking alcohol.

### Prevalence of sleep quality

The overall prevalence of poor sleep quality was 68.4%, at 95% CI (63.7, 72.7) with a mean GSQ score of 6.17 (SD ± 2.7). The average bed time of the participants was at 9:10 pm, whereas

**Table 1. Sociodemographic characteristics of study participants at Wadila primary hospital, Wadila, Northern Ethiopia, 2020.**

| | | Study groups n = 411 | |
|---|---|---|---|
| Variables | Category | Number of participants | Frequency % |
| Age | <20 | 52 | 12.7 |
| | 20–30 | 293 | 71.3 |
| | >30 | 66 | 16 |
| Educational status | No formal education | 123 | 29.9 |
| | Primary | 116 | 28.2 |
| | Secondary | 91 | 22.1 |
| | ≥Diploma | 81 | 19.8 |
| Marital status | Single | 14 | 3.4 |
| | Married | 390 | 94.9 |
| | Divorced | 0 | - |
| | Widowed | 7 | 1.7 |
| Occupation | Employed | 89 | 21.7 |
| | Merchant | 42 | 10.2 |
| | Farmer | 45 | 10.9 |
| | House wife | 205 | 50.4 |
| | Others* | 28 | 6.8 |
| Residence | Urban | 197 | 47.9 |
| | Rural | 214 | 52.1 |
| Monthly income (ETB) | <1000 | 154 | 37.5 |
| | 1000–4000 | 234 | 56.9 |
| | >4000 | 23 | 5.6 |

the average wake up time was 6:45 am. The mean total time to fall asleep was 42 minutes (SD + 27 minutes) and the actual sleep time ranges from 2 up to 10 hours with a mean of 6.7 hours (SD +1.6) as reported on the PSQI.

The seven components of sleep quality were assessed using the PSQI and accordingly only one (0.2%) participant rate her subjective sleep quality very bad and 87.1% of pregnant mothers had ≤7 hours of sleep per night. On the other hand, no participant reports the use of sleep medication during the last month (a month before data collection). Moreover, the habitual sleep efficiency was ≥85% in 19.2% and <65% in 17.5% of the study participants (Table 2).

### Factors independently associated with poor sleep quality

From the total variables included in the backward logistic regression model, three variables were found to be statistically significant ($p < 0.05$) with 36.7% fitness of model summary (Hosmer and Lemeshow test). Accordingly, age [age 20–30 years; AOR = 4.3 CI (1.8, 9.9), $p = 0.001$, and age >30 years; AOR = 4.7 CI (1.6, 13.9) $p = 0.005$], gestational age [second trimester, AOR = 2.46, CI (1.2, 4.9), p = 0.01 and third trimester, AOR = 7.5, CI (3.2, 17.8), p = 0.000] and parity [multiparous women; AOR = 2.1(1.24, 3.6) p = 0.006] were identified to have statistically significant association with poor sleep quality (Table 3).

### Discussion

Sleep is a physiological need for all human kind. Pregnant women, in particular, need adequate sleep to develop their fetuses as well as save energy required for delivery [8]. Pregnancy is a process that creates significant anatomical, physiological, and biochemical changes in a woman's life [4]. These changes affect the physical and emotional behaviours of women and may

**Table 2. Sleep quality and its component scores among study participants at Wadila primary hospital, Wadila, Northern Ethiopia, 2020.**

| Variables | Values | Number of participants | Frequency (%) |
|---|---|---|---|
| Study participants (n = 411) | | | |
| Sleep duration | >7 | 123 | 29.9 |
| | 6–7 | 187 | 45.5 |
| | 5–6 | 54 | 13.2 |
| | <5 | 74 | 11.4 |
| Sleep latency | Never (0) | 53 | 12.9 |
| | <1 time a week (1) | 121 | 29.4 |
| | 1–2 times a week (2) | 168 | 40.9 |
| | ≥3 times a week (3) | 69 | 16.8 |
| Sleep efficiency | ≥85% | 79 | 19.2 |
| | 75–84% | 157 | 38.2 |
| | 65–74% | 103 | 25.1 |
| | <65% | 72 | 17.5 |
| Day time dysfunction | Never (0) | 278 | 67.6 |
| | <1 time a week (1) | 125 | 30.4 |
| | 1–2 times a week (2) | 7 | 1.7 |
| | ≥3 times a week (3) | 1 | 0.2 |
| Sleep disturbance | Never (0) | 17 | 4.1 |
| | <1 time a week (1) | 376 | 91.5 |
| | 1–2 times a week (2) | 18 | 4.4 |
| | ≥3 times a week (3) | 0 | 0 |
| Use of sleep medication | Never (0) | 411 | 100 |
| | <1 time a week (1) | 0 | 0 |
| | 1–2 times a week (2) | 0 | 0 |
| | ≥3 times a week (3) | 0 | 0 |
| Subjective sleep quality | Very good (0) | 122 | 29.7 |
| | Fairly good (1) | 261 | 63.5 |
| | Fairly bad (2) | 27 | 6.6 |
| | Very bad (3) | 1 | 0.2 |

lead to sleep disturbances. Poor sleep quality can result in pre-term birth, gestational hypertension and cesarean section deliveries [15, 17].

The results of this study indicate that women experience significant sleep problems during pregnancy. One of the most common problems experienced by pregnant women was frequent night wakings (92.5%). From the total participants 68.6% of women experienced difficulty falling asleep within 30 minutes and waking to use toilet. More than a quarter (27.7%) of the women also found to have subjective snoring problem during pregnancy. These findings were supported by other studies conducted in Vietnam [18], China [17] and India [2].

On the other hand, 29.7% of pregnant mothers rate their sleep quality very good, 87.1% of study participants had ≤7 hours of sleep per night and no participant reports the use of sleep medication during the last month. More ever, the habitual sleep efficiency was ≥85% in 19.2% and <65% in 17.5% of the study participants. This finding is also similar to other studies conducted in Denmark [3], Iran [19], and Vietnam [18]. Mechanical factors like frequent urination, very active baby, heart burn and hormonal factors like, increased level of estrogen and oxytocin might be responsible for increased sleep disturbance during pregnancy [4–7].

**Table 3. Bivariable and multivariable logistic regression model of factors independently associated with poor sleep quality among pregnant mothers at Wadila primary hospital, Wadila, Northern Ethiopia, 2020, (N = 411).**

| Variables | Poor sleep quality | | OR (95% CI) | | | |
|---|---|---|---|---|---|---|
| | Yes N (%) | No N (%) | COR (95% CI) | p-value | AOR (95% CI) | p-value |
| Age | | | | | | |
| <20 | 22(42.3) | 30(57.7) | 1 | | 1 | |
| 20–29 | 204(69.6) | 89(30.4) | 3.13(1.7, 5.7) | <0.0001 | 4.3(1.8, 9.9) | 0.001 * |
| ≥30 | 55(83.3) | 11(16.7) | 6.8(2.9, 15.9) | <0.0001 | 4.7(1.6, 13.9) | 0.005 * |
| Monthly income | | | | | | |
| <1000 | 97(63) | 47(37) | 1.3(0.539,3.178) | 0.552 | 1.3(0.5, 3.6) | 0.642 |
| 1000–4000 | 171(73.1) | 63(26.9) | 2.1(0.872, 5.001) | 0.099 | 2.4(1.3, 6.2) | 0.066 |
| >4000 | 13(56.5) | 10(43.5) | 1 | | 1 | |
| Gestational age | | | | | | |
| First trimester | 25(50) | 25(50) | 1 | | 1 | |
| Second trimester | 146(65.5) | 77(34.5) | 1.9(1.021, 3.522) | 0.043 | 2.46(1.2, 4.9) | 0.010* |
| Third trimester | 110(79.7) | 28(20.3) | 3.9(1.966, 7.851) | <0.0001 | 7.5(3.2, 17.8) | <0.0001* |
| Parity | | | | | | |
| Nulliparous | 76(53.5) | 66(46.5) | 1 | | 1 | |
| Multiparous | 205(76.2) | 64(23.8) | 2.8(1.8, 4.289)) | <0.0001 | 2.1(1.24, 3.6) | 0.006* |

* Statistically significant at p-value <0.05; COR: Crude Odds Ratio; AOR: Adjusted Odds Ratio.

According to this study, the overall prevalence of poor sleep quality was 68.4%. This is in line with the studies conducted in China (64%) [20] and Malaysia (69.4%) [12]. However, our finding is lower than the findings of another similar study in USA (76%) [21], Iran (77%) [8], Indonesia (78.9%) [22], Turkey (86%) [13], and Korea (96.2%) [10]. The results of this study is higher than studies conducted in Brazil, 56.3% [23], Vietnam, 41.2% [18] and Pakistan, 53.3% [24]. The possible reason for this discrepancy may be the difference in sample size, study setting and socioeconomic status of the populations.

Maternal age was identified as a predictor of poor sleep quality in our study, suggesting that sleep quality of pregnant mothers declines as their age increases. This was in agreement with the results of studies conducted in China [17] and Turkey [13]. As a potential risk factor, women with increased age were more likely influenced by physiological and psychological factors, hence leading to the decline of sleep quality [17].

A statistically significant association between gestational age and poor sleep quality was observed among pregnants in this study, showing that sleep quality decreases as pregnancy continues. When compared with the first trimester of pregnancy, significant decline in sleep quality were observed in the second and third trimester of pregnancy. This is similar with the findings of studies conducted in Turkey [4], Pakistan [24] and China [17]. This might be due to the increment of hormonal changes, maternal stress and fetal movement as gestational age increases.

Another variable that predicts the occurrence of poor sleep quality among pregnant women in this study was parity. In the present study, multiparous women were 2.1 times more likely to have poor sleep quality than nulliparous women. This is in agreement with the result of the studies conducted in Vietnam [18] and USA [25]. The possible justification could be due to the fact that multiparous ladies are likely to have a young child in the home and their total sleep duration and sleep efficiency are likely to be influenced by childcare demands and children's sleep programs, including nighttime wakings [25].

### Limitation of the study

Due to the cross-sectional nature of the study, a causal relation cannot be established and sample size was limited.

### Conclusions

More than two-third of pregnant mothers had poor sleep quality. Advanced age, increased gestational age and multiparty are found to be determinants of poor sleep quality in pregnant women. Poor sleep quality may bring about the mother's mental impairment, and the offspring are predisposed to various anxiety disorders and learning disabilities. Therefore, pregnant mothers need to have health education about risk factors of poor sleep quality and sleep hygiene practice.

### Supporting information

**S1 Appendix. Amharic version questionnaire.**
(DOCX)

**S2 Appendix. English version questionnaire.**
(DOCX)

### Acknowledgments

We would like to express our appreciation to Debre Tabor University for its support to do this research project. We would like to thank our study participants and data collectors for their willingness and cooperation.

### Author Contributions

**Conceptualization:** Tadeg Jemere, Berhanu Getahun, Fitalew Tadele, Gashaw Walle.

**Data curation:** Berhanu Getahun, Belayneh Kefale, Gashaw Walle.

**Formal analysis:** Belayneh Kefale, Gashaw Walle.

**Investigation:** Tadeg Jemere, Berhanu Getahun, Belayneh Kefale.

**Methodology:** Tadeg Jemere, Berhanu Getahun.

**Project administration:** Tadeg Jemere, Fitalew Tadele.

**Software:** Belayneh Kefale.

**Supervision:** Fitalew Tadele, Belayneh Kefale.

**Visualization:** Tadeg Jemere, Fitalew Tadele.

**Writing – original draft:** Tadeg Jemere, Berhanu Getahun, Fitalew Tadele, Belayneh Kefale, Gashaw Walle.

**Writing – review & editing:** Tadeg Jemere, Berhanu Getahun, Fitalew Tadele, Belayneh Kefale, Gashaw Walle.

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
