## [Decision Letter · Decision Letter 0]

12 Feb 2021

PONE-D-20-34586

Poor sleep quality and its associated factors among pregnant women in Northern Ethiopia, 2020.

PLOS ONE

Dear Dr. Jemere,

Thank you for submitting your manuscript to PLOS ONE. After careful consideration, we feel that it has merit but does not fully meet PLOS ONE’s publication criteria as it currently stands. Therefore, we invite you to submit a revised version of the manuscript that addresses the points raised during the review process.

We look forward to receiving your revised manuscript.

Kind regards,

Nülüfer Erbil, Ph.D, Prof.

Academic Editor

PLOS ONE

Journal Requirements:

2. Please include additional information regarding the survey or questionnaire used in the study and ensure that you have provided sufficient details that others could replicate the analyses. For instance, if you developed a questionnaire as part of this study and it is not under a copyright more restrictive than CC-BY, please include a copy, in both the original language and English, as Supporting Information."

Furthermore, we note that you have reported significance probabilities of 0 in places. Since p=0 is not strictly possible, please correct this to a more appropriate limit, eg 'p<0.0001.

3. In the ethics statement in the manuscript and in the online submission form you have stated that IRB approval was obtained from ethical review Committee of Debre Tabor University, while data was collected from Wadila primary hospital. Please provide further clarification whether IRB of the participating institution also approved the study prior to data collection.

5. Thank you for submitting the above manuscript to PLOS ONE. During our internal evaluation of the manuscript, we found significant text overlap between your submission and the following previously published works.

https://www.oatext.com/pdf/NPC-2-152.pdf

https://bmcresnotes.biomedcentral.com/articles/10.1186/s13104-019-4531-6

https://www.sciencedirect.com/science/article/abs/pii/S2352721819300725?via%3Dihub

https://linkinghub.elsevier.com/retrieve/pii/S088421751533848X

Please revise the manuscript to rephrase the duplicated text, cite your sources, and provide details as to how the current manuscript advances on previous work. Please note that further consideration is dependent on the submission of a manuscript that addresses these concerns about the overlap in text with published work.

Reviewers' comments:

Reviewer's Responses to Questions

**Comments to the Author**

1. Is the manuscript technically sound, and do the data support the conclusions?

Reviewer #1: Yes

Reviewer #2: Yes

Reviewer #3: Yes

2. Has the statistical analysis been performed appropriately and rigorously? 

Reviewer #1: Yes

Reviewer #2: Yes

Reviewer #3: Yes

3. Have the authors made all data underlying the findings in their manuscript fully available?

Reviewer #1: Yes

Reviewer #2: Yes

Reviewer #3: No

4. Is the manuscript presented in an intelligible fashion and written in standard English?

Reviewer #1: Yes

Reviewer #2: Yes

Reviewer #3: Yes

5. Review Comments to the Author

Reviewer #1: Dear Author,

I think it is a very nice and important subject. However, it would be appropriate to make some adjustments about the content. You can find my suggestions below:

1. The introduction should be rearranged, first the importance of sleep, sleep changes during pregnancy, the reasons and consequences of this should be written.

2. Subheadings should be rearranged according to the journal's writing rules, and published studies should be reviewed.

3. Confusion in the method section should be removed and written in more understandable expressions.

4. An explanation should be added that the permission to use was obtained from the owner of the scale of PSQI.

5. In the first paragraph, only the importance of sleep should be mentioned and sleep disorders should not be mentioned (subjective snoring)

6. Humans or pregnants should be mentioned rather than “mammals”.

7. 3. paragraphs should be 2. Paragraphs, so a change of location should be made.

8. First, the population, the sample, then the inclusion and exclusion criteria should be written separately in the methods section.

9. “Pregnant mothers who use any of the substances such as alcohol, khat, and cigarette smoking at least once in their life time were classified as ever user.” Because of this sampling, this sentence is not required.

10. Was the 5% sample tested before the study included in the study? It should be written.

11. It is written as Study Groups: 198 in Table 1. Isn't the sample size 411?

12. In the discussin section, “…The possible reason for this discrepancy may be the difference in PSQI cut of point, sample size, study setting and socioeconomic status of the populations.”

a. Cut off point of PSQI must be constant. The fact that it differs according to the situation is not suitable for scale evaluation. This comment should be revised.

b. Spelling mistakes should be corrected (for example: cut off point)

13. In the limitation of study section,“Despite these limitations, our study clearly showed the magnitude of poor sleep quality and its predictors in our country, which is not well investigated so far.”

a. This is not a limitation statement.

14. After conclusion, a recommendation sentence for the study should be developed.

15. Sources dating back 10 years should not be used (like references 4 and 16).

Reviewer #2: This paper studies the relationship between the quality of sleep and its associated factors among pregnant mothers in Ethiopia. 411 patients data are used for the study and the results are analyzed using SPSS. The results show that two-third of patients have poor sleep quality that needs further care such as health and education. The paper is easy to follow.

The paper needs to consider the following concerns;

1. There are plenty of studies on the relationship between the quality of sleep and pregnancy. Therefore, the side effects of poor sleep quality could be better motivated.

2. The procedure of collecting and analyzing data could be better shown using a figure or flowchart.

3. The conclusion of the paper needs to include more detail regarding the study.

4. The sections should be numbered correctly in order to easily follow them. The paper organization can be added to the first section as well.

5. The paper needs a proofread for English problems.

Reviewer #3: the manuscript is new and will add value to the body of knowledge

edit conclusion to only include the findings in this study

There is no data provided other than the findings presented in table and words.

there is no fear of dual publication, however, references needs to be updated to less than 10 years old

6. PLOS authors have the option to publish the peer review history of their article (what does this mean?). If published, this will include your full peer review and any attached files.

Reviewer #1: No

Reviewer #2: No

Reviewer #3: **Yes: **Auma A G

---

## [Author Response · Author response to Decision Letter 0]

3 Mar 2021

Response to the editor

Manuscript title: Poor sleep quality and its associated factors among pregnant women in Northern Ethiopia, 2020.

Manuscript number: PONE-D-20-34586

Dear editor: Thank you for giving us the chance to revise the manuscript. Saying this we addressed all the concerns raised by the reviewers and incorporated the authors’ reflection in the revised manuscript. 

Editor’s comments

Author response: Appropriate modification was done. See the revised manuscript.

2. Please include additional information regarding the survey or questionnaire used in the study and ensure that you have provided sufficient details that others could replicate the analyses. For instance, if you developed a questionnaire as part of this study and it is not under a copyright more restrictive than CC-BY, please include a copy, in both the original language and English, as Supporting Information."

Furthermore, we note that you have reported significance probabilities of 0 in places. Since p=0 is not strictly possible, please correct this to a more appropriate limit, eg 'p<0.0001.

Author response: The questionnaire is attached as additional information both in original language and English. The other issues are corrected. See the revised manuscript.

3. In the ethics statement in the manuscript and in the online submission form you have stated that IRB approval was obtained from ethical review Committee of Debre Tabor University, while data was collected from Wadila primary hospital. Please provide further clarification whether IRB of the participating institution also approved the study prior to data collection.

Author response: First Ethical review Committee of Debre Tabor University approved the study then Debre Tabor University write letter of cooperation to conduct the study in Wadila primary hospital. Latter Ethical review Committee of Wadila primary hospital also approved the study and the study was conduct after the permission was obtained from the Medical director of the Hospital.

Author response: Corrected. See the revised manuscript.

5. Thank you for submitting the above manuscript to PLOS ONE. During our internal evaluation of the manuscript, we found significant text overlap between your submission and the following previously published works.

https://www.oatext.com/pdf/NPC-2-152.pdf

https://bmcresnotes.biomedcentral.com/articles/10.1186/s13104-019-4531-6

https://www.sciencedirect.com/science/article/abs/pii/S2352721819300725?via%3Dihub

https://linkinghub.elsevier.com/retrieve/pii/S088421751533848X

Please revise the manuscript to rephrase the duplicated text, cite your sources, and provide details as to how the current manuscript advances on previous work. Please note that further consideration is dependent on the submission of a manuscript that addresses these concerns about the overlap in text with published work.

Author response: Corrected. See the revised manuscript.

Point by point response for reviewers’ comments 

Manuscript title: Poor sleep quality and its associated factors among pregnant women in Northern Ethiopia, 2020.

Manuscript number: PONE-D-20-34586

Dear reviewers: Thank you for reviewing the manuscript. Saying this we addressed all the concerns raised by the reviewers and incorporated the authors’ reflection in the revised manuscript. 

Reviewer one comments/questions 

I think it is a very nice and important subject. However, it would be appropriate to make some adjustments about the content. You can find my suggestions below:

Authors’ Response: Thank you very much. We accept your comments and correct them all. See revised manuscript.

1. The introduction should be rearranged, first the importance of sleep, sleep changes during pregnancy, the reasons and consequences of this should be written. 

 Author response: Appropriate modification was done. See the revised manuscript.

2. Subheadings should be rearranged according to the journal's writing rules, and published studies should be reviewed.

Author response: Appropriate measure was taken. See the revised manuscript.

3. Confusion in the method section should be removed and written in more understandable expressions.

Author response: Thank you. Appropriate measure was taken. See the revised manuscript.

4. An explanation should be added that the permission to use was obtained from the 

owner of the scale of PSQI.

Author response: It is already adapted and widely used here in Ethiopia. 

5. In the first paragraph, only the importance of sleep should be mentioned and sleep 

 disorders should not be mentioned (subjective snoring)

Author response: Corrected. See the revised manuscript.

6. Humans or pregnants should be mentioned rather than “mammals”.

Author response: Corrected. See the revised manuscript.

7. 3. Paragraphs should be 2. Paragraphs, so a change of location should be made.

Author response: Appropriate modification was done. See the revised manuscript.

8. First, the population, the sample, and then the inclusion and exclusion criteria should be written separately in the methods section.

Author response: Corrected. See the revised manuscript.

9. “Pregnant mothers who use any of the substances such as alcohol, khat, and cigarette smoking at least once in their life time were classified as ever user.” Because of this sampling, this sentence is not required.

Author response: Removed. See the revised manuscript.

10. Was the 5% sample tested before the study included in the study? It should be written.

Author response: They were not included in the study. 

11. It is written as Study Groups: 198 in Table 1. Isn't the sample size 411?

Author response: Thank you very much! Corrected. See the revised manuscript.

12. In the discussion section, “…The possible reason for this discrepancy may be the difference in PSQI cut of point, sample size, study setting and socioeconomic status of the populations.” 

a. Cut off point of PSQI must be constant. The fact that it differs according to the situation is not suitable for scale evaluation. This comment should be revised.

b. Spelling mistakes should be corrected (for example: cut off point) 

Author response: Corrected. See the revised manuscript.

13. In the limitation of study section, “Despite these limitations, our study clearly showed the magnitude of poor sleep quality and its predictors in our country, which is not well investigated so far.”

a. This is not a limitation statement.

 Author response: Corrected. See the revised manuscript.

14. After conclusion, a recommendation sentence for the study should be developed.

Author response: Corrected. See the revised manuscript.

15. Sources dating back 10 years should not be used (like references 4 and 16).

Author response: Thank you very much! Reference 4 is replaced by recently published article, but we could not find updated article for reference 16. See the revised manuscript.

Response to reviewer #2 comments

Reviewer #2: This paper studies the relationship between the quality of sleep and its associated factors among pregnant mothers in Ethiopia. 411 patients data are used for the study and the results are analyzed using SPSS. The results show that two-third of patients have poor sleep quality that needs further care such as health and education. The paper is easy to follow. The paper needs to consider the following concerns;

1. There are plenty of studies on the relationship between the quality of sleep and pregnancy. Therefore, the side effects of poor sleep quality could be better motivated.

Author response: Off course there are plenty of studies on the relationship between the quality of sleep and pregnancy worldwide. But to our knowledge this is the first study in Ethiopia. So our aim was first to know the magnitude of poor sleep quality among pregnant mothers in Ethiopia, then we will continue to investigate its impact on the outcome of pregnancy in our next project.

2. The procedure of collecting and analyzing data could be better shown using a figure or flowchart.

Author response: It can be showed like that. But we choose this one due to the study design.

3. The conclusion of the paper needs to include more detail regarding the study.

Author response: Corrected. See the revised manuscript.

4. The sections should be numbered correctly in order to easily follow them. The paper organization can be added to the first section as well.

Author response: Corrected. See the revised manuscript.

5. The paper needs a proofread for English problems.

Author response: Appropriate modification was done. See the revised manuscript.

Response to Reviewer #3 comments

The manuscript is new and will add value to the body of knowledge. Edit conclusion to only include the findings in this study. There is no data provided other than the findings presented in table and words. There is no fear of dual publication, however, references needs to be updated to less than 10 years old.

Author response: Corrected. See the revised manuscript.

1. General Comment: It is a new addition to the body of knowledge; it is not so much studied and there is a lot of gray area to be explored. However, the title could include the study design too.

Author response: Corrected. See the revised manuscript.

2. Abstract : In your conclusion, you indicated that the offspring are predisposed to various anxiety disorders and learning disabilities as a result of poor sleep quality, is this from the study result? If so, please indicate how you measured it. Wadila primary hospital is a study site and is not necessary in the list of the key words.

Author response: It is not from the result of this study. Corrected. See the revised manuscript.

3. Introduction: Introduction is well stated, although will need to be aligned from global perspective, regional to the nation and the local back ground of the subject. Consider including relevant controversies around this field. Check if the aim of the study was achieved in this particular paper and include in the background statement.

Author response: Corrected. See the revised manuscript.

4. Methodology: Study method is well aligned, sample size adequate. Data collection procedure adequately explained. However, indicate how a systematic random sampling got you to 422 samples. Clearly state the statistical significance of the variables on the methodology.

Author response: Corrected. See the revised manuscript.

5. Discussion: The discussion is well brought out; however, it does don urge out what other studies have found, what it means to this study and why the similarities or differences.

Author response: Corrected. See the revised manuscript.

6. Conclusion/Recommendation: The conclusion suits the study findings, but stick to the findings only.

Author response: Corrected. See the revised manuscript.

7. References: Update all references older than 10 years to a more recent study. Some of your references are very old, number 1, 4, 10 and 16.

Author response: References 1 and 4 are replaced by recently published article, but we could not find updated article for reference 16. See the revised manuscript.

---

## [Decision Letter · Decision Letter 1]

19 Apr 2021

Poor sleep quality and its associated factors among pregnant women in Northern Ethiopia, 2020: A cross sectional study

PONE-D-20-34586R1

Dear Dr. Jemere,

We’re pleased to inform you that your manuscript has been judged scientifically suitable for publication and will be formally accepted for publication once it meets all outstanding technical requirements.

Kind regards,

Nülüfer Erbil, Ph.D, Prof.

Academic Editor

PLOS ONE

---

## [Editor Report · Acceptance letter]

23 Apr 2021

PONE-D-20-34586R1 

Poor sleep quality and its associated factors among pregnant women in Northern Ethiopia, 2020: A cross sectional study 

Dear Dr. Jemere:

I'm pleased to inform you that your manuscript has been deemed suitable for publication in PLOS ONE. Congratulations! Your manuscript is now with our production department. 

Kind regards, 

on behalf of

Dr. Nülüfer Erbil 

Academic Editor

PLOS ONE